# DriveTransformer: Unified Transformer for Scalable End-to-End Autonomous Driving

**Xiaosng Jia\*, Junqi You\*, Zhiyuan Zhang\*, Junchi Yan[†]**
Sch. of Computer Science & Sch. of Artificial Intelligence, Shanghai Jiao Tong University
\* Equal Contributions       [†] Correspondence Author
https://github.com/Thinklab-SJTU/DriveTransformer/

## Abstract

End-to-end autonomous driving (E2E-AD) has emerged as a trend in the field of autonomous driving, promising a data-driven, scalable approach to system design. However, existing E2E-AD methods usually adopt the sequential paradigm of perception-prediction-planning, which leads to cumulative errors and training instability. The manual ordering of tasks also limits the system's ability to leverage synergies between tasks (for example, planning-aware perception and game-theoretic interactive prediction and planning). Moreover, the dense BEV representation adopted by existing methods brings computational challenges for long-range perception and long-term temporal fusion. To address these challenges, we present **DriveTransformer**, a simplified E2E-AD framework for the ease of scaling up, characterized by three key features: *Task Parallelism* (All agent, map, and planning queries direct interact with each other at each block), *Sparse Representation* (Task queries direct interact with raw sensor features), and *Streaming Processing* (Task queries are stored and passed as history information). As a result, the new framework is composed of three unified operations: task self-attention, sensor cross-attention, temporal cross-attention, which significantly reduces the complexity of system and leads to better training stability. **DriveTransformer** achieves state-of-the-art performance in both simulated closed-loop benchmark Bench2Drive and real world open-loop benchmark nuScenes with high FPS.

## 1 Introduction

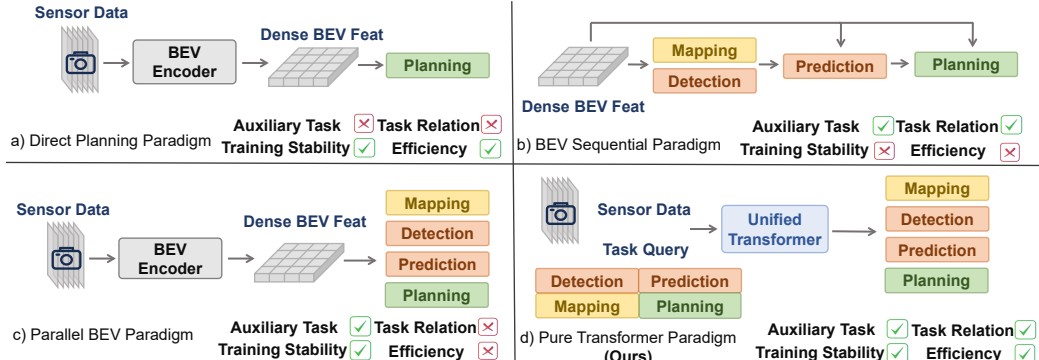

Figure 1: **End-to-End Autonomous Driving Paradigm Comparison.** The proposed pure Transformer paradigm avoids the construction of expensive BEV features and allows the tasks to learn their relations with raw sensor inputs, other tasks, and histories all by Transformer attention.

Autonomous driving has been a topic of interest (Li et al., 2023; Yang et al., 2023b) in recent years, with significant progress being made in the field (Hu et al., 2023; Jia et al., 2023b). One of

Correspondence author is also affiliated with Shanghai lnnovation Institute. This work was in part supported by NSFC (62222607) and Shanghai Municipal Science and Technology Major Project under Grant 2021SHZDZX0102.

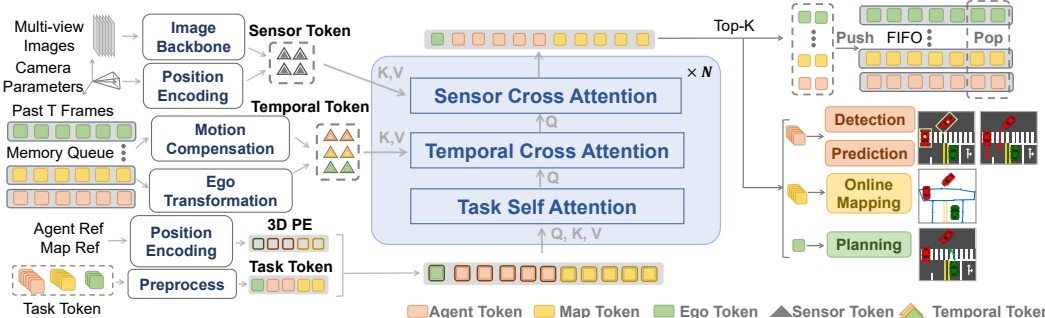

Figure 2: **Overall Framework of DriveTransformer.** DriveTransformer features streaming, parallel, and sparse token interaction. At each layer, task tokens interact with each other by *Task Self Attention*, extract information from raw sensor inputs by *Sensor Cross Attention*, and fuse temporal information from history task tokens in memory queue by *Temporal Cross Attention*.

the most exciting approaches is end-to-end autonomous driving (E2E-AD), which aims to integrate perception (Li et al., 2023), prediction (Jia et al., 2023a), and planning (Li et al., 2024a) into a single, holistic framework. **E2E-AD is particularly appealing due to its data-driven (Lu et al., 2024) and scalable nature, allowing for continuous improvement with more data**.

Despite these advantages, existing E2E-AD methods (Hu et al., 2023; Jiang et al., 2023) mostly adopt a sequential pipeline of perception-prediction-planning, where downstream tasks are heavily dependent on upstream queries. **This sequential design can lead to cumulative errors and thus training instability**. For instance, the training process of UniAD (Hu et al., 2023) necessitates a multi-stage approach: first, pre-training the BEVFormer encoder (Li et al., 2022b); then, training TrackFormer and MapFormer; and finally, training MotionFormer and Planner. This fragmented training approach increases the complexity and difficulty of deploying and scaling the system in industrial settings. Moreover, **the manual ordering of tasks can restrict the system's ability to leverage synergies**, such as planning-aware perception (Philion et al., 2020; Jia et al., 2023d) and game-theoretic interactive prediction and planning (Jia et al., 2021; Huang et al., 2023).

Another challenge existing methods grapple with is the spatial-temporal complexity of the real world. BEV-based representations (Li et al., 2022b) encounter computational challenges with long-range detection (Jiang et al., 2024) due to the dense nature of the BEV grid. Additionally, the image backbone of BEV methods is under-optimized due to weak gradient signals (Yang et al., 2023a), hindering their ability to scale. For temporal fusion, BEV-based methods typically store history BEV features for fusion, which is computationally extensive as well (Park et al., 2023). In summary, **BEV-based methods ignore the sparsity of 3D space and drop the task query of each frame**, which results in significant waste of computation and thus suffer from efficiency (Li et al., 2024c).

The latest work ParaDrive (Weng et al., 2024) tries to mitigate the instability issue by removing all tasks' connection. However, it still suffers from the expensive BEV representation and their experiments is limited to open-loop, which could not reflect actual planning ability (Li et al., 2024c).

To address these deficiencies, we introduce **DriveTransformer**, a framework for efficient and scalable end-to-end autonomous driving, featuring three key properties as shown in Fig. 2:

- **Task Parallelism**: All task queries directly interact with each other at each block, fostering cross-task knowledge transfer while maintaining system stability without explicit hierarchy.
- **Sparse Representation**: Task queries directly engage with raw sensor features, offering an efficient and direct means of information extraction, aligning with end-to-end optimization paradigm.
- **Streaming Processing**: Temporal fusion is achieved through a first-in-first-out queue that stores task queries in history and temporal cross attention, ensuring efficiency and feature reuse.

DriveTransformer offers a unified, parallel, and synergistic approach to E2E-AD, facilitating easier training and scalability. As a result, DriveTransformer achieves state-of-the-art closed-loop performance in Bench2Drive (Jia et al., 2024) under CARLA simulation and state-of-the-art open-loop planning performance on nuScenes (Caesar et al., 2020b) dataset.

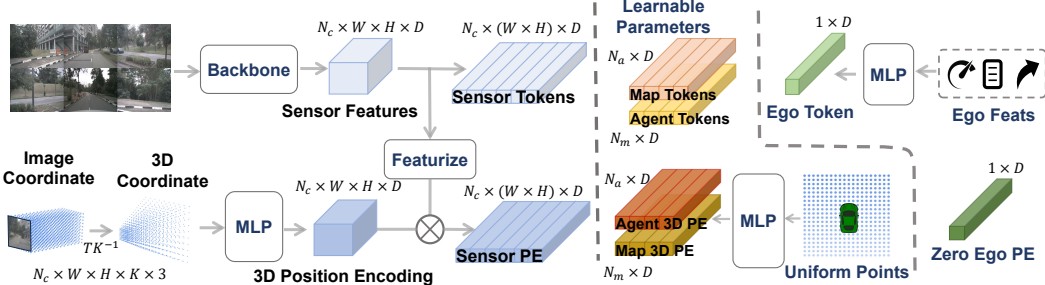

Figure 3: **Initialization & Tokenization Process.** Sensor inputs are processed by backbone while their PE are their 3D coordinate. Agent and Map tokens are initialized from learnable parameters while their initial PE are uniformly initialized. Ego token is initialized from canbus information while its PE is initialized as zeros.

## 2 RELATED WORKS

The concept of E2E-AD could date back to 1980s (Pomerleau, 1988). CIL (Codevilla et al., 2018) trained a simple CNN to map front-view camera images directly to control commands. Refined by CILRS (Codevilla et al., 2019), it incorporated an auxiliary task to predict the ego vehicle's speed, addressing issues related to inertia. PlanT (Renz et al., 2022) approach suggested leveraging a Transformer architecture for the teacher model, while LBC (Chen et al., 2020) focused on initially training a teacher model with privileged inputs. Moving forward, studies such as Zhang et al. (2021); Li et al. (2024a) ventured into reinforcement learning to create driving policies. Building on these advancements, student models were developed (Wu et al., 2022; Hu et al., 2022a). In subsequent research, the use of multiple sensors became prevalent, enhancing the models' capabilities. Transfuser (Prakash et al., 2021; Chitta et al., 2022) utilized a Transformer for the integration of camera and LiDAR data. LAV (Chen & Krähenbühl, 2022) adopted the PointPainting (Vora et al., 2020) technique, and Interfuser (Shao et al., 2022) incorporated safety-enhanced rules into the decision-making process. Further innovations included the usage of VectorNet for map encoding by MMFN (Zhang et al., 2022) and the introduction of a DETR-like scalable decoder paradigm by ThinkTwice (Jia et al., 2023d) for student models. ReasonNet (Shao et al., 2023) proposed specialized modules to improve the exploitation of temporal and global information, while Jaeger et al. (2023) suggested a classification-based approach to student's output to mitigate the averaging issue.

In another branch where AD sub-tasks are explicitly conducted, ST-P3 (Hu et al., 2022b) integrated detection, prediction, and planning tasks into a unified BEV segmentation framework. Further, UniAD (Hu et al., 2023) employed Transformer to link different tasks, and VAD (Jiang et al., 2023) proposed a vectorized representation space. ParaDrive (Weng et al., 2024) removes the links among all tasks while BEVPlanner (Li et al., 2024c) removes all middle tasks. Concurrent to our work, there are sparse query based methods (Zhang et al., 2024; Sun et al., 2024; Su et al., 2024). However, they still follow the sequential pipeline while the proposed DriveTransformer unifies all tasks into parallel Transformer paradigm.

## 3 METHOD

Given raw sensor inputs (e.g., multi-view images), **DriveTransformer** aims to output results for multiple tasks, including object detection (Li et al., 2022a), motion prediction (Jia et al., 2023c), online mapping (Chen et al., 2022), and planning (Li et al., 2024a). Each task is handled by its corresponding queries, which directly interact with each other, extract information from raw sensor inputs, and integrate information from histories. The overall framework is illustrated in Fig. 2.

### 3.1 INITIALIZATION & TOKENIZATION

Prior to information exchange in DriveTransformer, all inputs are converted into a unified representation - tokens. Inspired by DAB-DETR (Liu et al., 2022a), all tokens consist of two parts: **semantic embeddings** for semantic information and **position encodings** for spatial localization. In Fig. 3, we illustrate the process and we give details below.

**Sensor Tokens**: Multi-view images from surrounding cameras are separately encoded by backbones like ResNet (He et al., 2016) into $\boldsymbol{H}_{\text{sensor}} \in \mathbb{R}^{N_c \times H \times W \times D}$ semantic embeddings, where $N_c$ is the

**Sensor Cross Attention**     **Task Self-Attention**    **Temporal Cross Attention**

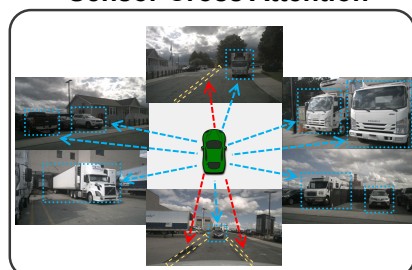 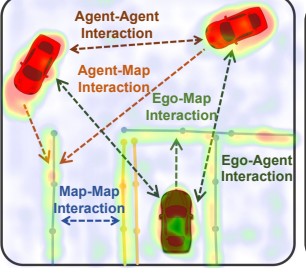 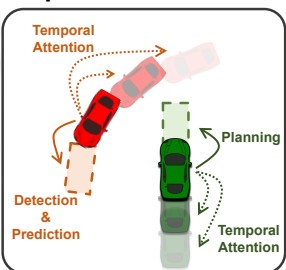

Figure 4: **Three Types of Attention in DriveTransformer.** Sensor Cross Attention provides a direct way for all tasks to access raw inputs in an end-to-end way. Task Self-Attention allows the information exchange among tasks. Temporal Cross Attention utilizes the history states as priors.

number of cameras, $H$ and $W$ are the height and width of the feature map after patchification, $D$ is the hidden dimension. To encode the position information of sensor features, we adopt 3D Position Encoding in (Liu et al., 2022b;c). Specifically, for each patch with pixel coordinate (i, j), its corresponding ray in the 3D space could be represented with $K$ equally spaced 3D points: $\mathbf{Ray}_{i,j} = \{TK^{-1}[i, j, d_k]|k = 1, 2, ..., K\}$, where $T$ and $K$ are the extrinsic and intrinsic matrix of the camera, $d_k$ is the depth value for the $k^{\text{th}}$ 3D point. Then, coordinates of points in the same ray are concatenated and fed into an MLP to obtain the position encoding for each patch (Liu et al., 2022c) and the 3D position encoding for all image patches is denoted as $\mathbf{PE}_{\text{sensor}} \in \mathbb{R}^{N_c \times H \times W \times D}$

**Task Tokens**: To model the heterogeneous driving scene, three types of **task queries** are initialized to extract different information: (I) **Agent Queries** represent dynamic objects (vehicles, pedestrians, etc), which will be used to conduct object detection and motion prediction. (II) **Map Queries** represent static elements (lanes, traffic signs, etc), which will be used to conduct online mapping. (III) **Ego Query** represents the potential behavior of the ego vehicle, which will be used to conduct planning. Following DAB-DETR (Liu et al., 2022a), both agent queries' and map queries' semantic embeddings are initialized randomly as learnable parameters $\boldsymbol{H}_{\text{agent}} \in \mathbb{R}^{N_a \times D}$ and $\boldsymbol{H}_{\text{map}} \in \mathbb{R}^{N_m \times D}$ where $N_a$ and $N_m$ are the number of agent and map queries - pre-defined hyper-parameters. Their position encodings $\mathbf{PE}_{\text{agent}} \in \mathbb{R}^{N_a \times D}$ and $\mathbf{PE}_{\text{map}} \in \mathbb{R}^{N_m \times D}$ are initialized uniformly within pre-defined perception range. For planning query, its semantic embedding is initialized from an MLP encoding canbus information $\boldsymbol{H}_{\text{ego}} = \text{MLP}(\boldsymbol{H}_{\text{canbus}}) \in \mathbb{R}^D$ similar to BEVFormer (Li et al., 2022b) while its position encoding $\mathbf{PE}_{\text{ego}} \in \mathbb{R}^D$ is initialized as all zeros.

### 3.2 TOKEN INTERACTION

**All information exchange within DriveTransformer is established by the vanilla attention mechanisms (Vaswani et al., 2017), ensuring scalability and easy deployment**. As a result, the model could be trained under one stage and demonstrate strong scalability, which will be shown in the experiments section. In following sub-sections, we describe the three types of information exchange adopted at each layer of DriveTransformer and the illustration is in Fig. 4

**Sensor Cross Attention (SCA)** establishes a direct pathway between tasks and raw sensor inputs, enabling end-to-end learning without information loss. SCA is conducted as:

$$\boldsymbol{H}'_{\text{ego}}, \boldsymbol{H}'_{\text{agent}}, \boldsymbol{H}'_{\text{map}} = \text{SCA-Attention}(Q = [\boldsymbol{H}_{\text{ego}} + \mathbf{PE}_{\text{ego}}, \boldsymbol{H}_{\text{agent}} + \mathbf{PE}_{\text{agent}}, \boldsymbol{H}_{\text{map}} + \mathbf{PE}_{\text{map}}],$$
$$K = \boldsymbol{H}_{\text{sensor}} + \mathbf{PE}_{\text{sensor}}, \quad V = \boldsymbol{H}_{\text{sensor}}) \quad (1)$$

where $\boldsymbol{H}'$ denotes the updated query. In this way, raw sensor tokens are matched by task queries based on both semantic and spatial relations to extract task-specific information in an end-to-end way without information loss. Notably, by adopting 3D position encoding (Liu et al., 2022c), **DriveTransformer avoids the construction of BEV feature, which is efficient and has less gradient vanishing issue (Yang et al., 2023a), allowing scaling up**.

**Task Self-Attention (TSA)** enables direct interaction among arbitrary tasks without explicit constraint, promoting synergy such as planning-aware perception (Philion et al., 2020) and game-theoretic interactive prediction and planning (Huang et al., 2023). TSA is conducted as:

$$\boldsymbol{H}'_{\text{ego}}, \boldsymbol{H}'_{\text{agent}}, \boldsymbol{H}'_{\text{map}} = \text{TSA-Attention}(Q = [\boldsymbol{H}_{\text{ego}} + \mathbf{PE}_{\text{ego}}, \boldsymbol{H}_{\text{agent}} + \mathbf{PE}_{\text{agent}}, \boldsymbol{H}_{\text{map}} + \mathbf{PE}_{\text{map}}],$$
$$K = [\boldsymbol{H}_{\text{ego}} + \mathbf{PE}_{\text{ego}}, \boldsymbol{H}_{\text{agent}} + \mathbf{PE}_{\text{agent}}, \boldsymbol{H}_{\text{map}} + \mathbf{PE}_{\text{map}}], \quad V = [\boldsymbol{H}_{\text{ego}}, \boldsymbol{H}_{\text{agent}}, \boldsymbol{H}_{\text{map}}]) \quad (2)$$

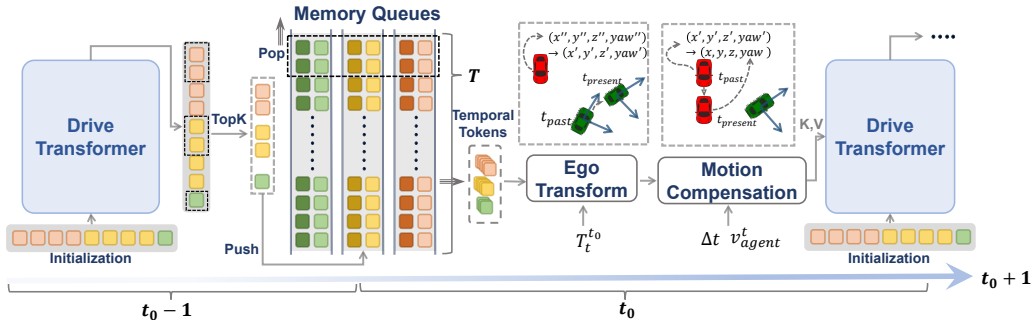

Figure 5: **Streaming Temporal Mechanism in DriveTransformer.** Top-K task queries at previous timestep from the last layer of DriveTransformer are pushed into FIFO queues. Task queries and positions are transformed into current ego coordinate system and are compensated for potential movement before feeding into Temporal Cross Attention as Key and Value.

By eliminating manually designed task dependencies, **the interleaved relations among tasks could be flexibly learnt via attention in a data-driven way**, which eases the scaling up. In contrast, UniAD (Hu et al., 2023) has to adopt a multi-stage training strategy due to the inconsistency at the early stage of training where inaccurate upstream modules influence downstream modules and finally collapse the whole training.

**Temporal Cross Attention** integrates information from previously observed history. Existing paradigms (Hu et al., 2023; Jiang et al., 2023) use history BEV features to pass temporal information, which introduces two drawbacks: (a) Maintaining long-term BEV features is expensive (Park et al., 2023) (b) Previous task queries carrying strong prior semantic and spatial information are wastefully discarded. Inspired by StreamPETR (Wang et al., 2023), DriveTransformer maintains First-In-First-Out (FIFO) queues of queries for each task respectively and conducts cross attention to history queries in the queue at each layer to fuse temporal information, as illustrated in Fig. 5.

Specifically, denote $\boldsymbol{H}_{\text{ego}}^t$, $\boldsymbol{H}_{\text{agent}}^t$, $\boldsymbol{H}_{\text{map}}^t$ with their corresponding position encodings $\mathbf{PE}_{\text{ego}}^t$, $\mathbf{PE}_{\text{agent}}^t$, $\mathbf{PE}_{\text{map}}^t$ as the ego queries, agent queries, and map queries of the final layer of DriveTransformer at time-step $t$. Suppose current time-step is $t_0$ and we maintain FIFO queues $\text{Queue}_{\text{ego}} = \{(\boldsymbol{H}_{\text{ego}}^t, \mathbf{PE}_{\text{ego}}^t)|t = t_0 - 1, t_0 - 2, ..., t_0 - T_{\text{queue}}\}$, $\text{Queue}_{\text{agent}} = \{(\boldsymbol{H}_{\text{agent}}^t, \mathbf{PE}_{\text{agent}}^t)|t = t_0 - 1, t_0 - 2, ..., t_0 - T_{\text{queue}}\}$ and $\text{Queue}_{\text{map}} = \{(\boldsymbol{H}_{\text{map}}^t, \mathbf{PE}_{\text{map}}^t)|t = t_0 - 1, t_0 - 2, ..., t_0 - T_{\text{queue}}\}$ where $T_{\text{queue}}$ is a pre-set hyper-parameter to control the length of temporal queue. After each time-step, current task queries at the final layer are pushed into the queue and the task queries at $t_0 - T$ are popped out. Further, since there are redundant queries in DETR style methods (Carion et al., 2020), for agent and map queries, only those with top-K confidence scores are kept, where K is a hyper-parameter.

Temporal Cross Attention uses the history queries as Key and Value. Since the ego reference point at different time-step could be different, the history queries' PE is transformed into current coordinate system (Ego Transformation):

$$\hat{\mathbf{PE}}^t = \text{MLP}(T_t^{t_0}\mathbf{Pos}^t) \quad \text{where } t = t_0 - 1, t_0 - 2, ..., t_0 - T_{\text{queue}} \tag{3}$$

where $\hat{\mathbf{PE}}^t$ is the transformed PE, $T_t^{t_0}$ is the coordinate transformation matrix from $t$ to $t_0$. Besides, since other agents could have their own movement, following (Wang et al., 2023), we conduct DiT (Peebles & Xie, 2022) style ada-LN for Motion Compensation:

$$\hat{\mathbf{PE}}_{\text{agent}}^t = \text{LayerNorm}(\mathbf{PE}_{\text{agent}}^{\hat{t}}, [\gamma, \beta] = \text{MLP}(v_{\text{agent}}^t * (t - t_0))) \text{ where } t = t_0 - 1, ..., t_0 - T_{\text{queue}} \tag{4}$$

where the layer-norm's weight $\gamma$ and bias $beta$ is controlled by the predicted velocity of agents at time-step $t$ and the time interval between $t$ and current time-step $t_0$. Besides, we also set the relative time embedding as $t_{\text{emb}} = \text{MLP}(t - t_0)$ to indicate different time-steps and Temporal Cross-Attention is conducted as:

$$\boldsymbol{H}_{\text{task}}' = \text{TCA-Attention}(Q = \boldsymbol{H}_{\text{task}} + \mathbf{PE}_{\text{task}}, \ K = \{\boldsymbol{H}_{\text{task}}^t + \hat{\mathbf{PE}}_{\text{task}}^t + t_{\text{emb}}|t = t_0 - 1, ..., t_0 - T_{\text{queue}}\},$$
$$V = \{\boldsymbol{H}_{\text{task}}^t|t = t_0 - 1, ..., t_0 - T_{\text{queue}}\}) \text{ where task=Ego, Map, Agent}$$
$$\tag{5}$$

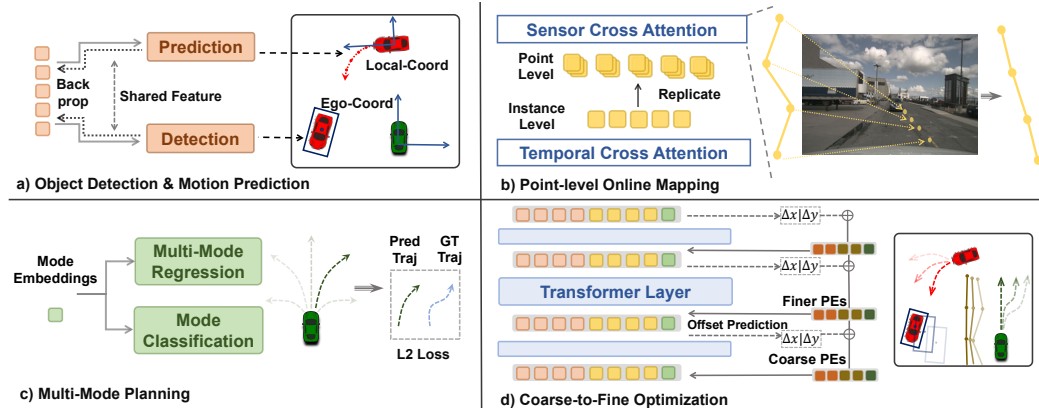

Figure 6: **Task Head Designs.** (a) Detection and Motion share the same agent feature, which associate objects without tracking. Trajectory prediction in local coordinate system further disentangles the two tasks, ensuring training stability. (b) Map points in the same polyline have different PEs in Sensor Cross Attention to retrieve fine-grained features. (c) Planning head combines the ego query with different mode embeddings to conduct multi-mode prediction. (d) Tasks' positions are gradually refined and their corresponding PEs are gradually more accurate, providing better interaction.

**Pure Attention Architecture: In summary, DriveTransformer is a stack of multiple blocks where each block contains the aforementioned three attentions and a FFN**:

$$\boldsymbol{H}_{\text{ego}}^{l+1}, \boldsymbol{H}_{\text{agent}}^{l+1}, \boldsymbol{H}_{\text{map}}^{l+1} = \text{FFN}(\text{TSA}((\text{TCA}(\text{SCA}([\boldsymbol{H}_{\text{ego}}^{l}, \boldsymbol{H}_{\text{agent}}^{l}, \boldsymbol{H}_{\text{map}}^{l}], \boldsymbol{H}_{\text{sensor}}), \boldsymbol{H}_{\text{task}}'),$$
$$[\text{Queue}_{\text{ego}}, \text{Queue}_{\text{agent}}, \text{Queue}_{\text{map}}])) \tag{6}$$

where $l$ and $l+1$ is the layer index, FFN means MLP in Transformer (Vaswani et al., 2017) and we omit the PE, residual connection, and pre-layernorm for brevity. Note that the raw sensor tokens $\boldsymbol{H}_{\text{sensor}}$ and history information $\text{Queue}_{\text{ego}}, \text{Queue}_{\text{agent}}, \text{Queue}_{\text{map}}$ are shared across all blocks.

## 3.3 DETR-Style Task Head

Inspired by DETR (Carion et al., 2020; Liu et al., 2022a), **task heads are set after each block to gradually refine the predictions and the PE would be correspondingly updated**. In following sub-sections, we introduce task specific designs and the updating strategy of PE, as shown in Fig. 6.

**Object Detection & Motion Prediction**: Existing E2E methods (Zeng et al., 2022; Hu et al., 2023) still adopt the classic detection-association-prediction pipeline, which introduces instability to training due to the inherent difficulty of association (Weng et al., 2022). For example, in UniAD, there must be a 3D object detection pretrained BEVFormer to avoid the divergence of TrackFormer and then the MapFormer and TrackFormer must be primarily trained before the end-to-end training of MotionFormer and planning head, which necessitates its multi-stage training strategy and thus hinders the scaling up.

To alleviate this issue, DriveTransformer adopts a more end-to-end methodology: **conducting object detection and motion prediction without tracking, by feeding the same agent query into different task heads**. The same feature for the same agent naturally establish associations between detection and prediction. For temporal association, since Temporal Cross Attention is conducted between current tokens and *all* history tokens, the explicit association is avoided, replaced by the learning-based attention mechanism. To further improve the training stability and reduce the interference between these two tasks, the label of motion prediction is transformed to the local coordinate system of each agent (Jia et al., 2022) and thus its loss is not influenced by detection results at all. Only during inference, the waypoints from the prediction are transformed into the global coordinate system based on the detection to calculate motion prediction related metrics.

**Online Mapping** Recent progress in the sub-field (Li et al., 2024b; Liu et al., 2024) suggests the importance of point-level instead of instance-level feature retrieval due to the irregular and diverse distribution of map polylines. Thus, when conducting *Sensor Cross Attention*, we replicate each map query for $N_{\text{point}}$ times paired with position encodings for each single point. In this way, for

Table 1: **Planning Performance in Bench2Drive**. Avg. L2 is averaged over the predictions in 2 seconds under 2Hz. * denotes expert feature distillation. All latency are measured by the averaged inference step-time on CARLA evaluation in A6000.

| Method | Open-loop Metric | Closed-loop Metric | | | | Latency |
|---|---|---|---|---|---|---|
| | Avg. L2 ↓ | Driving Score ↑ | Success Rate(%) ↑ | Efficiency ↑ | Comfortness ↑ | |
| AD-MLP (Zhai et al., 2023) | 3.64 | 18.05 | 0.00 | 48.45 | 22.63 | **3ms** |
| UniAD-Tiny (Hu et al., 2023) | 0.80 | 40.73 | 13.18 | 123.92 | 47.04 | 420.4ms |
| UniAD-Base (Hu et al., 2023) | 0.73 | 45.81 | 16.36 | 129.21 | 43.58 | 663.4ms |
| VAD (Jiang et al., 2023) | 0.91 | 42.35 | 15.00 | **157.94** | **46.01** | 278.3ms |
| DriveTransformer-Large (**Ours**) | **0.62** | **63.46** | **35.01** | 100.64 | 20.78 | 211.7ms |
| TCP* (Wu et al., 2022) | 1.70 | 40.70 | 15.00 | 54.26 | 47.80 | **83ms** |
| TCP-ctrl* | - | 30.47 | 7.27 | 55.97 | **51.51** | 83ms |
| TCP-traj* | 1.70 | 59.90 | 30.00 | 76.54 | 18.08 | 83ms |
| TCP-traj w/o distillation | 1.96 | 49.30 | 20.45 | **78.78** | 22.96 | 83ms |
| ThinkTwice* (Jia et al., 2023d) | **0.95** | 62.44 | 31.23 | 69.33 | 16.22 | 762ms |
| DriveAdapter* (Jia et al., 2023b) | 1.01 | **64.22** | **33.08** | 70.22 | 16.01 | 931ms |

those long polylines, **each point could retrieve raw sensor information with better locality**. To integrate separate point-level map queries into instance-level ones for other modules, we adopt a light-weight PointNet (Qi et al., 2017) with max-pooling and MLPs.

**Planning**: We model the the future movement of the ego vehicle as Gaussian Mixture Model to avoid mode averaging, as widely done in the motion prediction field (Liang et al., 2020). Specifically, we divide all training trajectories according to their direction and distance into six categories: Go Straight, Stop, Left Turn, Sharp Left Turn, Right Turn, Sharp Right Turn. To generate trajectories of these modes, six mode embeddings are generated by feeding their sine&cosine encoded position (Vaswani et al., 2017) into an MLP and then we add them to the ego feature to predict six mode-specific ego trajectories. During training, only the Ground-Truth mode's trajectory would be used to calculate regression loss, i.e. winner-take-all (Liang et al., 2020), and we also train a classification head to predict current mode. During inference, the trajectory from the mode with the highest confidence score would be used to compute metrics or execute.

**Coarse-to-Fine Optimization**: The success of DETR series (Carion et al., 2020) demonstrates the power of end-to-end learning by coarse-to-fine optimization. In DriveTransformer, the Position Encodings (PE) of all task queries are updated after each block based on current predictions. Specifically, **the Map and Agent PE are encoded by their corresponding predicted positions and semantic classes with an MLP to capture the spatial and semantic relations among elements**. The ego PE is encoded by **the predicted planning trajectory with an MLP to capture the ego intentions for the possible interactions**. Similar to DETR, During training, losses are applied on task heads at all blocks while during inference we only use the output from the last block.

### 3.4 LOSS & OPTIMIZATION

**DriveTransformer is trained under one single stage**, where each tasks could gradually learn to find out their relations in Task Self-Attention, without collapsing each others' basic convergence under Sensor Cross Attention and Temporal Cross Attention. There are detection loss (DETR-style hungarian matching loss (Carion et al., 2020)), prediction loss (winner-take-all style loss (Liang et al., 2020)), online mapping loss (MapTR (Liao et al., 2023) style hungarian matching loss), and planning loss (winner-take-all style loss) where we adjust weights to make sure all terms have the same magnitude around 1, as in the following equation:

$$\mathcal{L}_{\text{overall}} = w_{\text{detection}}\mathcal{L}_{\text{detection}} + w_{\text{motion}}\mathcal{L}_{\text{motion}} + w_{\text{mapping}}\mathcal{L}_{\text{mapping}} + w_{\text{planning}}\mathcal{L}_{\text{planning}} \quad (7)$$

## 4 EXPERIMENTS

### 4.1 DATASET & BENCHMARK

We use Bench2Drive (Jia et al., 2024), a closed-loop evaluation protocol under CARLA Leaderboard 2.0 for end-to-end autonomous driving. It provides an official training set, where we use the base set (1000 clips) for fair comparison with all the other baselines. We use the official 220 routes for evaluation. Additionally, we compare our method with other state-of-the-art baselines on nuScenes (Caesar et al., 2020a) open-loop evaluation. There are three different size of models:

When comparing with SOTA works, we report results of DriveTransformer-Large. For ablation studies, since evaluating on 220 routes of Bench2Drive could take days, we select 10 representa-

Table 2: **Multi-Ability Results of E2E-AD Methods.** * denotes expert feature distillation.

| Method | Ability (%) ↑ | | | | | |
| --- | --- | --- | --- | --- | --- | --- |
| | Merging | Overtaking | Emergency Brake | Give Way | Traffic Sign | **Mean** |
| AD-MLP (Zhai et al., 2023) | 0.00 | 0.00 | 0.00 | 0.00 | 0.00 | 0.00 |
| UniAD-Tiny (Hu et al., 2023) | 7.04 | 10.00 | 21.82 | 20.00 | 14.61 | 14.69 |
| UniAD-Base (Hu et al., 2023) | 12.16 | 20.00 | 23.64 | 10.00 | 13.89 | 15.94 |
| VAD (Jiang et al., 2023) | 7.14 | 20.00 | 16.36 | 20.00 | 20.22 | 16.75 |
| DriveTransformer-Large (**Ours**) | 17.57 | 35.00 | 48.36 | 40.00 | 52.10 | **38.60** |
| TCP* (Wu et al., 2022) | 17.50 | 13.63 | 20.00 | 10.00 | 6.81 | 13.59 |
| TCP-ctrl* | 9.23 | 5.00 | 9.10 | 10.00 | 6.81 | 8.03 |
| TCP-traj* | 12.50 | 22.73 | 52.72 | 40.00 | 46.63 | 34.92 |
| TCP-traj w/o distillation | 14.71 | 7.50 | 38.18 | 50.00 | 29.97 | 28.03 |
| ThinkTwice* (Jia et al., 2023d) | 13.72 | 22.93 | 52.99 | 50.00 | 47.78 | 37.48 |
| DriveAdapter* (Jia et al., 2023b) | 14.55 | 22.61 | 54.04 | 50.00 | 50.45 | **38.33** |

Table 3: **Open-loop planning performance in nuScenes under VAD metric**. [†] denotes LiDAR-based methods. [‡] denotes the usage of ego-status.

| Method | L2 (m) ↓ | | | | Collision (%) ↓ | | | |
| --- | --- | --- | --- | --- | --- | --- | --- | --- |
| | 1s | 2s | 3s | Avg. | 1s | 2s | 3s | Avg. |
| NMP (Zeng et al., 2019)[†] | - | - | 2.31 | - | - | - | 1.92 | - |
| SA-NMP (Zeng et al., 2019)[†] | - | - | 2.05 | - | - | - | 1.59 | - |
| FF (Hu et al., 2021)[†] | 0.55 | 1.20 | 2.54 | 1.43 | 0.06 | 0.17 | 1.07 | 0.43 |
| EO (Khurana et al., 2022)[†] | 0.67 | 1.36 | 2.78 | 1.60 | 0.04 | 0.09 | 0.88 | 0.33 |
| ST-P3 (Hu et al., 2022b) | 1.33 | 2.11 | 2.90 | 2.11 | 0.23 | 0.62 | 1.27 | 0.71 |
| UniAD (Hu et al., 2023) | 0.48 | 0.74 | 1.07 | 0.76 | 0.12 | 0.13 | 0.28 | 0.17 |
| VAD-Tiny (Jiang et al., 2023) | 0.46 | 0.76 | 1.12 | 0.78 | 0.21 | 0.35 | 0.58 | 0.38 |
| VAD-Base (Jiang et al., 2023) | 0.41 | 0.70 | 1.05 | 0.72 | 0.07 | 0.17 | 0.41 | 0.22 |
| BEVPlaner (Li et al., 2024c) | 0.27 | 0.54 | 0.90 | 0.57 | - | - | - | - |
| DriveTransformer-Large (**Ours**) | 0.19 | 0.34 | 0.66 | 0.40 | 0.03 | 0.10 | 0.21 | 0.11 |
| VAD-Tiny[‡] (Jiang et al., 2023) | 0.20 | 0.38 | 0.65 | 0.41 | 0.10 | 0.12 | 0.27 | 0.16 |
| VAD-Base[‡] (Jiang et al., 2023) | 0.17 | 0.34 | 0.60 | 0.37 | 0.07 | 0.10 | 0.24 | 0.14 |
| AD-MLP[‡] (Zhai et al., 2023) | 0.20 | **0.26** | **0.41** | **0.29** | 0.17 | 0.18 | 0.24 | 0.19 |
| BEVPlaner++[‡] (Li et al., 2024c) | **0.16** | 0.32 | 0.57 | 0.35 | - | - | - | - |
| ParaDrive[‡] (Weng et al., 2024) | 0.25 | 0.46 | 0.74 | 0.48 | 0.14 | 0.23 | 0.39 | 0.25 |
| DriveTransformer-Large[‡] (**Ours**) | **0.16** | 0.30 | 0.55 | 0.33 | **0.01** | **0.06** | **0.15** | **0.07** |

Table 4: **Size Configuration of DriveTransformer.** Latency is measured by the averaged model time for closed-loop evaluation in CARLA. Training batch size is measured by A800 (80G) to fill the GPU memory. Driving Score is under Dev10 benchmark.

| Size | Configuration | #Parameters | Latency | Training Batch Size | Driving Score |
| --- | --- | --- | --- | --- | --- |
| Small | 3 Layers, 256 Hidden | 47.41M | 93.8ms | 48 | 45.04 |
| Base | 6 Layers, 512 Hidden | 178.05M | 139.6ms | 28 | 60.45 |
| Large | 12 Layers, 768 Hidden | 646.33M | 221.6ms | 12 | 68.22 |

tive scenes (namely **Dev10**) balancing behaviors weathers, and towns and report results on it with DriveTransformer-base for quick validation.

### 4.2 COMPARISON WITH STATE-OF-THE-ART METHODS

We compare DriveTransformer with SOTA E2E-AD methods in Table 1, Table 2, and Table 3. We observe that DriveTransformer persistently outperforms SOTA methods. From Table 1, **DriveTransformer has a lower inference latency compared to UniAD and VAD**. Notably, because of the unified, sparse, and streaming Transformer design, DriveTransformer could be trained with batch size 12 in A800 (80G) while UniAD with batch size 1 and VAD with batch size 4.

### 4.3 ABLATION STUDIES

In ablation studies, all closed-loop experiments are conducted on **Dev10**, a subset of Bench2Drive 220 routes, and all open-loop results are on Bench2Drive official validation set (50 clips). Please

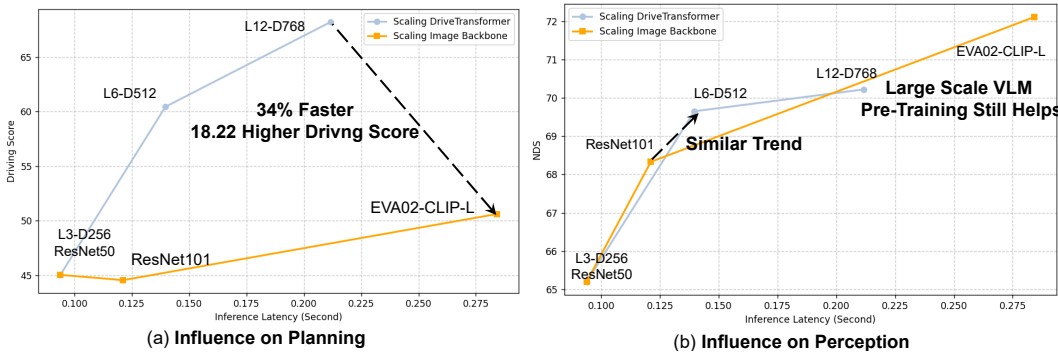

(a) **Influence on Planning** (b) **Influence on Perception**

Figure 7: **Scaling Study of Driving Transformer in Dev10 benchmark.** Directly scaling up the unified Transformer structure benefits most on the planning while adopting powerful image backbones, especially large-scale pretrained ones brings gains for perception.

Table 5: **Paradigm Design Study with DriveTransformer-Base in Dev10.**

(a) **Attention Mechanism**

| Method | Driving Score ↑ | Success Rate ↑ |
|---|---|---|
| Full Attention | **60.45** | **30.00** |
| w/o Sensor-CA | 8.41 | 0.00 |
| w/o Task-SA | 52.37 | 20.00 |
| w/o Temporal-CA | 56.22 | 20.00 |

(b) **Training Strategies.**

| Method | Driving Score ↑ | Success Rate ↑ |
|---|---|---|
| DriveTransformer | **60.45** | **30.00** |
| Planning Only | 54.22 | 20.00 |
| Pretrain Perception | **60.22** | **30.00** |
| w/o Middle Supervision | 51.67 | 10.00 |

Table 6: **Task Head Design Study with DriveTransformer-base.**

(a) **Detection & Prediction**

| Method | minADE ↓ |
|---|---|
| Local Prediction | **1.34** |
| Global Prediction | 2.68 |

(b) **Mapping.**

| Method | mAP ↑ |
|---|---|
| Point PE | **20.25** |
| Line PE | 14.55 |

(c) **Planning.**

| Method | Driving Score ↑ | Success Rate ↑ |
|---|---|---|
| Multiple Mode | **60.45** | **30.00** |
| Single Mode | 49.19 | 20.00 |

check Appendix B for details. We use a smaller model (6 layers and 512 hidden dimension) for ablation studies to save computational resource if not specified. We will open source **Dev10** protocol, model code, and model checkpoints.

**Scaling Study**: One attractive characteristic of Transformer-based paradigm is its extremely strong scalability (Radford et al., 2019; Brown et al., 2020). Since DriveTransformer is composed of Transformers, we study the scaling behavior of increasing layers and hidden dimension simultaneously and compare the strategy with scaling image backbones similar to (Hu et al., 2023; Jiang et al., 2023) as shown in Fig. 7. We could observe that scaling up the decoder part, i.e., the number of layers and width for the three attention brings more gains compared to scale up image backbones. It is natural since the former one directly adds more computation to the planning task. On the other hand, for the perception task, scaling up the decoder has similar trend with scaling up image backbones, which demonstrates the generalizing ability of the proposed DriveTransformer. However, it still falls behind the large scale vision-language pretraining image backbone - EVA02-CLIP-L (Fang et al., 2024), aligning with findings in (Wang et al., 2023). It could be an important direction to study how to combine VLLM with autonomous driving (Yang et al., 2023b).

**Paradigm Design Study**: In Table 5, we ablate the design of DriveTransformer. We conclude that: ❶ Based on Table 5a, **all three types of attention are helpful**. ❷ It makes sense that Sensor Cross Attention is especially important since model would drive blindly without sensor information. ❸Temporal information has the least influence, which aligns with the findings in (Chitta et al., 2022). ❹Task Self-Attention could improve the driving score since the ego query could utilize the detected objects and map elements to conduct planning. ❺ Based on Table 5b, we could find that discarding auxiliary tasks leads to performance decay, which may come from the fact that it is rather difficult to fit the single planning output with high-dimensional inputs (surrounding camera images). Actually, it is a common practice in the end-to-end autonomous driving community to adopt auxiliary tasks to regularize the learned representations (Chen & Krähenbühl, 2022; Prakash et al., 2021; Jia et al., 2023d). ❻ **One stage training is enough** for convergence and the perception pretraining, i.e, two stage training, does not provide advantages. It comes from the design that there is no manual dependency among tasks and thus all tasks could first learn from the raw sensor

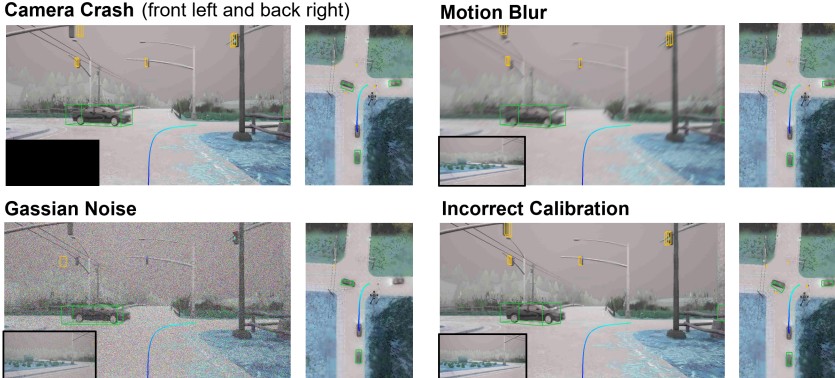

Figure 8: **Visualization of Detection and Planning under Different Robust Challenges.**

Table 7: **Robustness on Planning (Closed-Loop Evaluation)**. Driving Score↑ is reported.

| Methods | Regular | Camera Crash | Incorrect Calibration | Motion Blur | Gaussian Noise |
|---|---|---|---|---|---|
| VAD-Base (Jiang et al., 2023) | 53.45 | 48.54 (9.2%↓) | 38.46 (28.04%↓) | 45.47 (14.93%↓) | 44.53 (16.72%↓) |
| DriveTransformer (Ours) | 60.45 | 58.67 (**2.9%**↓) | 56.53 (**5.94%**↓) | 54.04 (**10.60%**↓) | 56.94 (**6.02%**↓) |

Table 8: **Robustness on Perception (Open-Loop Evaluation)**. Object detection NDS↑ is reported.

| Methods | Regular | Camera Crash | Incorrect Calibration | Motion Blur | Gaussian Noise |
|---|---|---|---|---|---|
| VAD-Base (Jiang et al., 2023) | 53.21 | 39.31(26.12%↓) | 43.01 (21.05%↓) | 42.34 (22.31%↓) | 45.16 (17.01%↓) |
| DriveTransformer (Ours) | 69.65 | 61.84 (**11.2%**↓) | 66.06(**5.15%**↓) | 61.61(**11.54%**↓) | 63.20(**9.26%**↓) |

inputs and history information instead of influencing each others' convergence. ❼ As a complex Transformer based framework, we find that the removal of supervision for middle layers collapse the training. It might need to further explore how to scale up the structure with only final supervisions.

**Task Design Study**: In Table 6, we ablate the designs of task heads and conclude that: ❶ For Table 6a, Formulating the prediction output in the local coordinate system decouples the object detection and motion prediction. As a result, the two tasks could optimize their objectives separately while the shared input agent features naturally associate the same agent. The superior performance demonstrates the effectiveness to avoid directly predict in the global coordinate system, aligns with (Jia et al., 2023c; Shi et al., 2024). ❷ For Table 6b, Point level PE in Sensor Cross Attention leads to significantly better online mapping results, which shows the effectiveness of extending the potential perception range for lane detection (Liu et al., 2024; Li et al., 2024b). ❸ For Table 6c, multi-mode planning outperforms single mode planning explicitly. By visualization, We observe that multi-mode planning achieves better control especially on scenarios requiring subtle steers. It comes from the fact that single mode prediction with L2 loss models the output space as one single Gaussian distribution and thus suffers from mode averaging.

### 4.4 ROBUSTNESS ANALYSIS

Autonomous driving, as an outdoor task, would frequently encounter many kinds of events and failures and thus it is an important perspective to examine the robustness of the system. To this end, We adopt 4 settings in (Xie et al., 2023). ❶ Camera Crash. Two cameras are masked as all black. ❷ Incorrect Calibration. rotation and transition noises are added to camera extrinsic parameters. ❸ Motion Blur is applied on images. ❹ Gaussian Noise is applied on images. From Table 7 and Table 8, DriveTransformer demonstrates significantly better robustness compared to VAD. It might be because VAD requires the construction of BEV feature, which is sensitive to perception inputs. On the other hand, **DriveTransformer directly interacts with raw sensor features and thus be able to ignore those failure or noisy inputs and demonstrates better robustness**.

### 5 CONCLUSION

In this work, we present DriveTransformer, a unified Transformer based paradigm for end-to-end autonomous driving, featured by task parallel, streaming processing, and sparse representation. It achieves state-of-the-art performance on both Bench2Drive in CARLA closed-loop evaluation and nuScenes open-loop evaluation with high FPS, demonstrating the efficiency of those designs.

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

## A  Implementation Details

We implement the model with Pytorch. When comparing with state-of-the-art works, we report results of DriveTransformer-Large. When conducting ablation studies, we report results of DriveTransformer-Base on Dev10 benchmark if not specified. All models are trained in Bench2Drive (Jia et al., 2024) base set (1000 clips) for 30 epochs on 8*A800 with a learning rate 1e-4, weight decay 0.05, dropout 0.1, AdamW, and cosine annealing schedule. We use ResNet50 as image backbones and the image size of (384, 1056) The temporal length ($T_{queue}$) are set as 10 (1 second 10Hz) in Bench2Drive and 4 (2 second 2Hz) in nuScenes. At each time-step, the Top-K queries is pushed into queue, where we set K as 50. The initial number of queries is set as 900 for agent queries following (Wang et al., 2023) and 100 for map queries following (Jiang et al., 2023). **We will open source our code and checkpoints**.

## B  Dev10 Benchmark

Bench2Drive (Jia et al., 2024) is a closed-loop evaluation protocol with 220 routes. The abundant number of routes could lower the variance of evaluation while introducing significant computational challenges. For example, it could take 2-3 days to evaluate DriveTransformer-Large on 8*A800. To this end, for fast development of ideas, we propose **Dev10** Benchmark by the following procedure:

1. There are 44 kinds of scenarios in Benc2Drive where each type has 5 different routes under different locations and weathers. Due to the low variance of Bench2Drive's short routes, selecting one route per scenario could reflect the model's ability in that case.

2. Further, there are some very similar scenarios in these 44 types. For example, as shown in page 15 of Bench2Drive's paper (https://arxiv.org/pdf/2406.03877), the difference between scnenario 2 "ParkingCutIn", scenario 3 "ParkingCutIn", and scenario 4 "StaticCutIn" all examine the ability of the ego vehicle to slow down or brake for the cut-in vehicle. Thus, after discussing with the Bench2Drive official team, these scenarios could be summarized into 10 high-level types:

- **ParkingExit**: requiring the model to drive out of a parking lot.
- **ParkingCrossingPedestrian**, DynamicObjectCrossing, ControlLoss, PedestrainCrossing, VehicleTurningRoutePedestrian, VehicleTurningRoute, HardBrake, OppositeVehicleRunningRedLight, OppositeVehicleTakingPriority: requiring the model to conduct emergency brake or slow down drastically under dangerous situations.
- **StaticCutIn**, HighwayExit, InvadingTurn, ParkingCutIn, HighwayCutIn: requiring the model to handle cut-in behaviors of the front vehicle*
- **HazardAtSideLane**, ParkedObstacle, Construction, Accident: requiring the model to overtake the blocking obstacles in front of it.
- **YieldToEmergencyVehicle**: requiring the model to give way to emergency vehicles.
- **ConstructionObstacleTwoWays**, ParkedObstacleTwoWays, AccidentTwoWays, VehiclesDooropenTwoWays, HazardAtSideLaneTwoWays: requiring the model to drive in the reverse lane for a short distance, complete the overtaking, and then return to the original lane.
- **NonSignalizedJunctionLeftTurn**, SignalizedJunctionLeftTurn, InterurbanActorFlow, InterurbanAdvancedActorFlow, CrossingBicycleFlow, VinillaNonSignalizedTurn, VinillaNonSignalizedTurnEncounterStopsign, VinillaSignalizedTurnEncounterGreenLight, VinillaSignalizedTurnEncounterRedLight, TJunction: requiring the model to handle the traffic at intersections and complete its turn.
- **BlockedIntersection**: requiring the model to yield for the blocking event within the intersection until the event is finished.
- **SequentialLaneChange**: requiring the model to continuously change several lanes.

- **SignalizedJunctionLeftTurnEnterFlow**, EnterActorFlows, SignalizedJunctionRight-Turn, NonSignalizedJunctionRightTurn, MergerIntoSlowTraffic, MergerIntoSlowTrafficV2: requiring the model to enter the dense traffic flow and merge into it.

3. For the 10 high-level types, we select one route for each with diverse weathers and towns. We give the details of Dev10 below:

Table 9: **Routes of Dev10 protocol.**

| Scenario | Route-ID | Road-ID | Town |
|---|---|---|---|
| ParkingExit | 3514 | 892 | 13 |
| ParkingCrossingPedestrian | 3255 | 1237 | 13 |
| StaticCutIn | 26405 | 137 | 15 |
| HazardAtSideLane | 25381 | 37 | 05 |
| YieldToEmergencyVehicle | 25378 | - | 03 |
| ConstructionObstacleTwoWays | 25424 | 269 | 11 |
| NonSignalizedJunctionLeftTurn | 2091 | - | 12 |
| BlockedIntersection | 27494 | 16 | 04 |
| SequentialLaneChange | 16569 | 1157 | 12 |
| SignalizedJunctionLeftTurnEnterFlow | 28198 | 234 | 15 |

4. We verify the variance of Dev10 with DriveTransformer-Base with 3 different seeds. The driving scores are 60.45, 59.20, 58.99 and the success rates 0.3, 0.3, 0.3, demonstrating very low variance.

5. When comparing with other methods, we stick to 220 routes to ensure fairness. When conducting ablation studies, we use Dev10 to save computational resource.

6. When comparing with other methods, we stick to 220 routes while conducting ablation studies, we use Dev10. Additionally, Dev10 could also serve as a validation set to avoid researchers overfitting Bench2Drive220. It could also avoid the overfit of Bench2Drive220 so that it could be a test set. We will open source the Dev10 benchmark and the Bench2Drive official team plans to integrate it into their official repo to provide a short and economic validation set.

**We will open source Dev10 protocol**.

## C  LIMITATIONS

Similar to existing end-to-end autonomous driving systems, DriveTrasnformer entangles the update of all sub-tasks and thus brings challenges for the maintenance of the whole system. An important future direction would make them less coupled and thus easier to debug and main separately.

Table 10: **Comparison of Performance on Middle Tasks.** † ParaDrive's latency is calculated by their claim that *2.77x speed up compared to UniAD* when disabling all middle tasks.

| Method | Detection | | Motion | | | Online Mapping | | Latency |
|---|---|---|---|---|---|---|---|---|
| | NDS | mAP | minADE | minFDE | MR | IoU-Road | IoU-Lane | |
| UniAD Hu et al. (2023) | 49.8 | 38.0 | 0.72 | 1.05 | 0.15 | 0.30 | 0.67 | 663.4ms |
| ParaDrive Weng et al. (2024) | 48.0 | 37.0 | 0.72 | - | - | 0.33 | 0.71 | 239.5ms† |
| **DriveTransformer** | **59.3** | **49.9** | **0.61** | **0.95** | **0.13** | **0.39** | **0.77** | **211.7ms** |

## D  COMPARISON OF MIDDLE TASKS

We compare the performance of middle tasks in nuScenes validation set as in Table 10.

## E  COMPARISON WITH CONCURRENT PARALLEL AND SPARSE BASED METHODS

Concurrent to DriveTransformer, there are other sparse-based methods including SparseAD Zhang et al. (2024) and SparseDrive Sun et al. (2024). To provide a comprehensive overview of existing sparse based methods, we compare DriveTransformer with them together with ParaDrive Weng et al. (2024), a most recent efficient modular E2E-AD method:

| Method | L2 (m) | | | | Collision (%) | | | | Latency |
|---|---|---|---|---|---|---|---|---|---|
| | 1s | 2s | 3s | Avg. | 1s | 2s | 3s | Avg. | |
| ParaDrive | 0.25 | 0.46 | 0.74 | 0.48 | 0.14 | 0.23 | 0.39 | 0.25 | 239.5ms |
| SparseAD-B | 0.15 | 0.31 | 0.57 | 0.35 | 0.00 | 0.06 | 0.21 | 0.09 | 285.7ms |
| SparseAD-L | 0.15 | 0.31 | 0.56 | 0.34 | 0.00 | 0.04 | 0.15 | 0.06 | 1428.6ms |
| SparseDrive-S | 0.29 | 0.58 | 0.96 | 0.61 | 0.01 | 0.05 | 0.18 | 0.08 | 111.1ms |
| SparseDrive-B | 0.29 | 0.55 | 0.91 | 0.58 | 0.01 | 0.02 | 0.13 | 0.06 | 137.0ms |
| DriveTransformer-S | 0.19 | 0.33 | 0.66 | 0.39 | 0.01 | 0.07 | 0.21 | 0.10 | 93.8ms |
| DriveTransformer-B | 0.16 | 0.31 | 0.56 | 0.34 | 0.01 | 0.06 | 0.16 | 0.08 | 139.6ms |
| DriveTransformer-L | 0.16 | 0.30 | 0.55 | 0.33 | 0.01 | 0.06 | 0.15 | 0.07 | 221.7ms |

We could observe that DriveTransformer achieves good L2 with high efficiency.

## F  TRAINING STABILITY & MULTI-STAGE TRAINING

In pioneering work UniAD Hu et al. (2023), they adopt a three step training strategy: (1) Training a BEVFormer Li et al. (2022b) with 3D object detection task; (2) Training TrackFormer and Map-Former; (3) Training all modules together. They explain in their paper that "*We first jointly train perception parts, i.e., the tracking and mapping modules, for a few epochs (6 in our experiments), and then train the model end-to-end for 20 epochs with all perception, prediction and planning modules. The two-stage training is found more stable empirically.*" and "*Joint learning. UniAD is trained in two stages which we find more stable.*".

We conduct experiments to train UniAD with one single stage (loading pretrained BEVFormer) in nuScenes shows that the overall loss (around 54) stops decreasing in early epoch (around epoch 4) while UniAD trained with the official two stage training has the final loss of (around 34), as shwon in Table 11.

Table 11: **Comparison of One-Stage and Two-Stage Trainging for UniAD.**

| Method | Final Loss | NDS (Detection) | IoU-lane (Mapping) | AMOTA(Tracking) | minADE(Motion) | Avg. L2 (Planning) |
|---|---|---|---|---|---|---|
| One-Stage | 54.07 | 38.1 | 23.5 | 20.4 | 2.52 | 2.01 |
| Two-Stage | **34.21** | **49.8** | **30.2** | **35.9** | **0.71** | **1.03** |

We could observe that one-stages result in underfit of all modules with much higher final loss.

Further, in concurrent work SparseAD Zhang et al. (2024), they mention that *The training is divided into **three stages in total**...stage 1 and stage 2 can be merged during training for a shorter training time with slight performance degradation in exchange.*. In SparseDrive Sun et al. (2024), they mention that *The training of SparseDrive is divided into **two stages**. In stage-1, we train symmetric sparse perception module from scratch to learn the sparse scene representation. In stage-2, sparse perception module and parallel motion planner are trained together.*

In contrast, compared to the sequential designs (UniAD, SparseAD, SparseDrive), one significant improvement in DriveTransformer is that **all tasks' interaction is learnt via attention instead of manual ordering**. As a result, at the early stage of training, each task could access information directly thorough sensor cross attention and temporal self attention, reducing the reliance on other under-trained tasks' queries. Such designs are friendly to the scaling up and industrial application of E2E-AD methods. As shown in Table 5, *Pretrain Perception* does not provide gains for Drive-Transformer, which proves the training stability of DriveTransformer.

