# OpenReview forum: "DriveTransformer: Unified Transformer for Scalable End-to-End Autonomous Driving"
_ICLR.cc/2025/Conference — ICLR 2025 Poster_

### Official Review · Reviewer_Fh94 · 2024-11-03

**Soundness:** 3
**Presentation:** 3
**Contribution:** 3
**Rating:** 8
**Confidence:** 4

**Summary:**

This paper proposes a new architecture for end-to-end driving, with three key features: task parallelism, sparse representation, streaming processing. With these key features, the proposed method reduces the complexity of the system and leads to better training stability, achieving SOTA performance on both open-loop and close-loop benchmarks.

**Strengths:**

1. It is a simple and effective idea using a vanilla transformer to unify sensor tokens and task queries.
2. Benefiting from the simple design, the model can scale up to achieve better performance.
3. The performance on close-loop benchmark is remarkable, surpassing previous methods by a large margin.
4. The writing and figures are easy to follow and understand.

**Weaknesses:**

1. The open-loop experiment should exclude ego-status information to prevent short-cut learning on nuScenes.
2. There seems no evidence to support that BEV Sequential Paradigm has bad training stability.

**Questions:**

1. Curious about the scaling ability, the performance seems not saturate, have you tried on larger model or larger data (Bench2Drive full dataset)?

---

> ### Author Response · Authors · 2024-11-20
>
> > **Q1: The open-loop experiment should exclude ego-status information to prevent short-cut learning on nuScenes.**
>
> Thanks for your kind advice! We have add the experiments of DriveTransformer-Large excluding ego-status information in Table 3 of [our updated paper](https://openreview.net/pdf?id=M42KR4W9P5). We could observe that DriveTransformer demonstrate good performance among methods excluding ego-status information.
>
> > **Q2: There seems no evidence to support that BEV Sequential Paradigm has bad training stability.**
>
> First, we would like to clarify that the instability is caused by *sequential designs*, as mentioned in the Introduction.
>
> The following is the evidence:
>
> - The cumulative errors and training instability is initially mentioned in UniAD(https://arxiv.org/pdf/2212.10156): Page 5: *We first jointly train perception parts, i.e., the tracking and mapping modules, for a few epochs (6 in our experiments), and then train the model end-to-end for 20 epochs with all perception, prediction and planning modules. The two-stage training is found more stable empirically.* Page 18: *Joint learning. UniAD is trained in two stages which we find more stable.* As in a discussions of meeting in CVPR, the authors of
> - In SparseAD(https://arxiv.org/pdf/2404.06892), they mention that: Page 30-31: *The training is divided into three stages in total...stage 1 and stage 2 can be merged during training for a shorter training time with slight performance degradation in exchange.*
> - In SparseDrive(https://arxiv.org/pdf/2405.19620), they mention that: Page 6-7: *The training of SparseDrive is divided into two stages. In stage-1, we train symmetric sparse perception module from scratch to learn the sparse scene representation. In stage-2, sparse perception module and parallel motion planner are trained together*.
>
> Our experiment with UniAD trained with one stage in nuScenes shows that the overall loss (around 54) stops decreasing in early epoch (around epoch 4) while UniAD trained with the official two stage training has the final loss of (around 34):
> | Method    | Final Loss | NDS (Detection) | IoU-lane (Mapping) | AMOTA(Tracking) | minADE(Motion) | Avg. L2 (Planning) |
> |-----------|------------|-----------------|------------|-----------------|----------------|--------------------|
> | One-Stage | 54.07      | 38.1            | 23.5       | 20.4            | 2.52           | 2.01               |
> | Two-Stage | **34.21**      | **49.8**            | **30.2**       | **35.9**            | **0.71**           | **1.03 *              |
>
> while as shown in Table 5 of the paper, *Pretrain Perception* does not provide gains for DriveTransformer.
>
> **Compared to the sequential designs (UniAD, SparseAD, SparseDrive), one significant improvement in DriveTransformer is that all tasks' interaction is learnt via attention instead of manual ordering**. As a result, at the early stage of training, each task could access information directly thorough sensor cross attention and temporal self attention, reducing the reliance on other under-trained tasks' queries. Such designs are friendly to the scaling up and industrial application of E2E-AD methods.
>
> Thanks for your advice to make the paper clearer and we have added the above description and experiment to Appendix F of [our updated paper](https://openreview.net/pdf?id=M42KR4W9P5) and marked the changes in blue.
>
>
> > **Q3: Curious about the scaling ability, the performance seems not saturate, have you tried on larger model or larger data (Bench2Drive full dataset)?**
>
> We share the same excitement that having bigger models may lead to better performance! Unfortunately, it is computationally forbiddable for us to train a larger models. However, **we will open source all our codes and hopefully members in the community would try to further scale up and check the effectiveness.**

---

> > ### Comment · Reviewer_Fh94 · 2024-11-26
> >
> > Thanks for your detailed response, that addressed my concerns and I will have my rating unchanged.

---

> ### Author Response · Authors · 2024-11-26
>
> Thanks for your reply and your highly positive rating. We are glad that your concerns are solved and we appreciate your advice to make the manuscript better!

---

### Official Review · Reviewer_6PT4 · 2024-11-03

**Soundness:** 3
**Presentation:** 3
**Contribution:** 2
**Rating:** 6
**Confidence:** 4

**Summary:**

This paper presents DriveTransformer, a unified Transformer framework for end-to-end autonomous driving. It features three key properties: task parallelism, sparse representation, and streaming processing, and contains three unified operations: task self-attention, sensor cross-attention, and temporal cross-attention, reducing system complexity and improving training stability. The framework achieves good performance in both the simulated closed-loop Bench2Drive and the open-loop nuScenes benchmarks.

**Strengths:**

● Originality: Proposes a new unified Transformer framework, combining sparse representation with parallelized task processing, distinct from existing methods.
● Quality: The design of each module is reasonable and effectively addresses existing problems.
● Clarity: The paper has a clear structure and it is easy to understand the core ideas.
● Significance: Provides new directions and ideas for autonomous driving research.

**Weaknesses:**

● Originality: Lacking enough original design. Because the task parallel concept is not entirely new (ParaDrive), and the sparse representation (core part following PETR & PETRv2) and streaming process lack strong originality.
● Comparisons: Lacking comparisons with similar methods in sparse and parallel aspects.

**Questions:**

1. In Figure 1, could it be explained why the training stability of b) is poor? And compared to c), why is the task relation of d) okay?
2. The streaming process maintains a FIFO process, retaining and reusing historical information. The BEV temporal fusion also maintains and reuses a fixed-length bev features. If it is a recurrent model, only the previous and next two frames are needed. So the BEV temporal processing of traditional methods can also be regarded as a streaming process. What is special about the streaming process in this paper?
3. In this paper, when comparing performance, training efficiency and inference latency, the comparisons are all made with UniAD and VAD. Is it possible to make a comparison at the same sparse method level with Sparsedrive and SparseAD, also at the same task parallelism method such as ParaDrive?
4. From Table 6c and Table 8, DriveTransformer shows significantly better robustness compared to VAD. Should Table 6c be Table 7?
5. When scaling up in this paper, the largest model used is 0.6b. Would the effect be better if the model were larger?
6. It is mentioned many times in the paper that the sequential model design can cause cumulative errors and training instability. Is it possible to make some comparisons between parallel and sequential methods, especially in terms of cumulative errors and training stability?

---

> ### Author Response · Authors · 2024-11-20
>
> Thanks for your acknowledgement and kind advice. Regarding your concerns, we give responses below:
>
> > **Q1:  Lacking enough original design. Because the task parallel concept is not entirely new (ParaDrive), and the sparse representation (core part following PETR & PETRv2) and streaming process lack strong originality.**
>
> Thanks for your opinion and we totaly understand your point. We agree that ParaDrive is an important work in the field since it points out that the manual ordering of task interaction in UniAD is sub-optimal. PETR, PETRv2, StreamPETR are indeed important inspirations of DriveTransformer.
>
> However, we argue that DriveTransformer has its own contributions that:
> - We highly admire the concept of 3D Position Encodings (3DPEs) and streaming in the PETR series. **Our unique contribution is that we formulated all tasks of AD and all interactions among tasks, sensors, sequences into pure Transformer + 3DPEs**. As a result, the proposed end-to-end autonomous driving (E2E-AD) system is highly scalable and efficient. We believe that our contributions from the E2E-AD perspective are insightful for the community.
> -  We agree that ParaDrive is earlier and we are exciting to know that there are similar findings: the manual ordering of task interaction in UniAD is sub-optimal. **We argue that our design is innovative compared to ParaDrive because ParaDrive proposes to simply remove all interactions among tasks while DriveTransformer proposes to achieve task interactions via attention.** In Table 5 (a), *w/o Task-SA* is similar to ParaDrive:
>
> | Task Interaction | Driving Score | Success Rate |
> | -------- | -------- | -------- |
> | No Task Interactions (ParaDrive Style)     | 52.37     | 20.00     |
> | Task Self-Attention (DriveTransformer) | **60.45** | **30.00** |
>
> We could observe that allowing task interactions are very helpful. It makes sense since there are indeed priors and relations among the diverse AD related tasks designed by pioneers of autonomous driving, which could speed up convergence and avoid overfitting.
>
>
> > **Q2: Comparisons: Lacking comparisons with similar methods in sparse and parallel aspects.**
>
> Thanks for your suggestion. We agree that ParaDrive, SparseAD, and SparseDrive are very important works in the field and comparing with them could make the community be clearer. we have added the comparision to [our updated paper](https://openreview.net/pdf?id=M42KR4W9P5) in Appendix E:
> | Method             | L2(1s) | L2(2s) | L2(3s) | L2(avg) | Col(1s) | Col(2s) | Col(3s) | Col(avg) | Latency  |
> |--------------------|--------|--------|--------|---------|---------|---------|---------|----------|----------|
> | ParaDrive          | 0.25   | 0.46   | 0.74   | 0.48    | 0.14    | 0.23    | 0.39    | 0.25     | 239.5ms  |
> | SparseAD-B         | 0.15   | 0.31   | 0.57   | 0.35    | 0.00    | 0.06    | 0.21    | 0.09     | 285.7ms  |
> | SparseAD-L         | 0.15   | 0.31   | 0.56   | 0.34    | 0.00    | 0.04    | 0.15    | 0.06     | 1428.6ms |
> | SparseDrive-S      | 0.29   | 0.58   | 0.96   | 0.61    | 0.01    | 0.05    | 0.18    | 0.08     | 111.1ms  |
> | SparseDrive-B      | 0.29   | 0.55   | 0.91   | 0.58    | 0.01    | 0.02    | 0.13    | 0.06     | 137.0ms  |
> | DriveTransformer-S | 0.19   | 0.33   | 0.66   | 0.39    | 0.01    | 0.07    | 0.21    | 0.10     | 93.8ms   |
> | DriveTransformer-B | 0.16   | 0.31   | 0.56   | 0.34    | 0.01    | 0.06    | 0.16    | 0.08     | 139.6ms  |
> | DriveTransformer-L | 0.16   | 0.30   | 0.55   | 0.33    | 0.01    | 0.06    | 0.15    | 0.07     | 221.7ms  |
>
> We could observe that DriveTransformer achieves good L2 with high efficiency. **As SparseAD and SparseDrive are concurrent works, we humbly request AC and reviewers that DriveTransformer should not be punished for frankly discussing about concurrent works**.

---

> > ### Author Response · Authors · 2024-11-20
> >
> > > **Q3: In Figure 1, could it be explained why the training stability of b) is poor? Is it possible to make some comparisons between parallel and sequential methods, especially in terms of cumulative errors and training stability?**
> >
> > - The cumulative errors and training instability is initially mentioned in UniAD(https://arxiv.org/pdf/2212.10156): Page 5: *We first jointly train perception parts, i.e., the tracking and mapping modules, for a few epochs (6 in our experiments), and then train the model end-to-end for 20 epochs with all perception, prediction and planning modules. The two-stage training is found more stable empirically.* Page 18: *Joint learning. UniAD is trained in two stages which we find more stable.* As in a discussions of meeting in CVPR, the authors of
> > - In SparseAD(https://arxiv.org/pdf/2404.06892), they mention that: Page 30-31: *The training is divided into three stages in total...stage 1 and stage 2 can be merged during training for a shorter training time with slight performance degradation in exchange.*
> > - In SparseDrive(https://arxiv.org/pdf/2405.19620), they mention that: Page 6-7: *The training of SparseDrive is divided into two stages. In stage-1, we train symmetric sparse perception module from scratch to learn the sparse scene representation. In stage-2, sparse perception module and parallel motion planner are trained together*.
> >
> > Our experiment with UniAD trained with one stage in nuScenes shows that the overall loss (around 54) stops decreasing in early epoch (around epoch 4) while UniAD trained with the official two stage training has the final loss of (around 34):
> > | Method    | Final Loss | NDS (Detection) | IoU-lane (Mapping) | AMOTA(Tracking) | minADE(Motion) | Avg. L2 (Planning) |
> > |-----------|------------|-----------------|------------|-----------------|----------------|--------------------|
> > | One-Stage | 54.07      | 38.1            | 23.5       | 20.4            | 2.52           | 2.01               |
> > | Two-Stage | **34.21**      | **49.8**            | **30.2**       | **35.9**            | **0.71**           | **1.03**            |
> >
> > while **as shown in Table 5 of the paper, *Pretrain Perception* does not provide gains for DriveTransformer**.
> >
> > **Compared to the sequential designs (UniAD, SparseAD, SparseDrive), one significant improvement in DriveTransformer is that all tasks' interaction is learnt via attention instead of manual ordering**. As a result, at the early stage of training, each task could access information directly thorough sensor cross attention and temporal self attention, reducing the reliance on other under-trained tasks' queries. Such designs are friendly to the scaling up and industrial application of E2E-AD methods.
> >
> > Thanks for your advice to make the paper clearer and we have added the above description and experiment to Appendix F of [our updated paper](https://openreview.net/pdf?id=M42KR4W9P5) and marked the changes in blue.
> >
> >
> >
> > > **Q4: In Figure 1, compared to c), why is the task relation of d) okay?**
> >
> > c) represents ParaDrive's solutions for the instability of sequential designs: they propose to **remove all interactions among tasks** and only allow task heads to directly interact with BEV features. In contrast, DriveTransformer proposes to let all tasks to learn their relationship via attention, a more learning way instead of manual designs. As a result, when the quality of other queries are low, they could always not to attend to them while when there is useful information, they have the ability to attend to them.
> >
> > > **Q5: So the BEV temporal processing of traditional methods can also be regarded as a streaming process. What is special about the streaming process in this paper?**
> >
> > As in L79-86 of the paper, there are two major benefits of adopting the proposed streaming framework:
> > - **Efficiency**: BEV features are expensive. For a +-51.2m x +-51.2m range and 0.512mx0.512m resolution in BEVFormer-base, it needs to store 200x200=40000 queries for each frame. In DriveTransformer, it needs to store 50 agent queries, 50 map queries, and 1 planning query at each frame, resulting 400x reducing of storing. What's worse, in the actual applications, the needs are around 8-16 frames in history, 150-200 meters in range, and resolutions less than 0.1x0.1m, which makes the sparse query based streaming more advantageous.
> > - **Priors**: The task queries are directly supervised by labels and thus stores rich semantic and spatial information while BEV features are more indirect. BEV feature based streaming frameworks fail to utilize task queries as history information, which is wasteful.

---

> ### Author Response · Authors · 2024-11-20
>
> > **Q6: From Table 6c and Table 8, DriveTransformer shows significantly better robustness compared to VAD. Should Table 6c be Table 7?**
>
> Thanks for your careful reading. Indeed, it is a typo. We have revised it.
>
> >**Q7: When scaling up in this paper, the largest model used is 0.6b. Would the effect be better if the model were larger?**
>
> We share the same excitement that having bigger models may lead to better performance! Unfortunately, it is computationally forbiddable for us to train a larger models. However, **we will open source all our codes and hopefully members in the community would try to further scale up and check the effectiveness.**

---

> > ### Comment · Reviewer_6PT4 · 2024-11-25
> >
> > Thanks for tha authors detailed responses, also the Associate Program Chairs' comments on my questions, which makes my comments/concerns more actionable. These responses addressed my concerns and questions, I believe this is a good paper with contribution to the community.

---

> > > ### Author Response · Authors · 2024-11-25
> > >
> > > Thanks for your reply. We are glad that your coners are solved and we appreciate your advice to make the manuscript better!

---

### Official Review · Reviewer_VVtW · 2024-11-03

**Soundness:** 2
**Presentation:** 2
**Contribution:** 2
**Rating:** 6
**Confidence:** 4

**Summary:**

This paper proposes a new architecture for end to end autonomous driving where all tasks are connected with others. Task queries and raw sensor features are connected. Also, the task queries are passed as history information for temporal cross attention. Experiments show competitive results with other SOTA architectures in the field.

**Strengths:**

The paper proposes a new architecture where all modules interact with each other. This should be able to capture complicated spatio-temporal dependencies among modules. They have also suggested using sensor cross attention and temporal cross attention, this can, at least in theory improve the generalization performance with sensor noise and failures or modules.

**Weaknesses:**

* The motivation for this architecture is not clear. When all modules interact with each other, we ignore the benefits of using modular end to end architectures. This would make the system difficult to debug and maintain. This can only be justified if the performance is significantly better than the modular architectures, however this does not seem to be the case.
* The results for detection, prediction, mapping is not shown. End to end driving is primarily about planning, however, it makes it more convincing if results of related tasks are also shown to be better.
* It would have useful to see the performance of ParaDrive in closed loop setting, as this architecture is closest in concept with ParaDrive. Only open loop comparison is shown.
* This manuscript could use a grammar / spell check, there are quite a few typos.

**Questions:**

* Task parallelism is not the right way to describe this architecture, what is parallel here?
* The benefit of this architecture needs to be motivated better, open loop results are not considerably better. Closed loop metrics are also mixed. The latency is also comparable. So, what is the benefit of using this architecture?
* What is the criterion for Dev10? 10 scenes out of 220 seem very selective, the criterion needs to be justified, otherwise the results could be prone to selection bias.

---

> ### Author Response · Authors · 2024-11-20
>
> Thanks for your acknowledgement and kind advice. Regarding your concerns, we give responses below:
>
> > **Q1: The motivation for this architecture is not clear.  When all modules interact with each other, we ignore the benefits of using modular end to end architectures. This would make the system difficult to debug and maintain.**
>
> Thanks for the question.
>
> - First, we would like to clarify that **DriveTransformer is a modular system** as well since it outputs the results of each sub-task. Thus, people could always debug and maintain according to the failure of each task (missing objects, missing centerlines, etc).
>
> - Second, as for the design of "all modules interact with each other", we agree that it would make the update of each module entangled. However, under existing end-to-end models (such as UniAD), all tasks have direct interactions with either BEV features or image features. Thus, updating one module in these models would also influence other modules.
>
> - In fact, current E2E-AD systems are all such double-edged swords: **to enjoy the benefits of raw features access for planning, the update of the whole system is entangled.**. This concern is actually an important future direction of E2E-AD community. We appreciate your point and have added it to the limitation section in the Appendix C of [our updated paper](https://openreview.net/pdf?id=M42KR4W9P5): *Similar to existing end-to-end autonomous driving systems, DriveTrasnformer entangles the update of all sub-tasks and thus brings challenges for the maintenance of the whole system. An important future direction would make them less coupled and thus easier to debug and main separately.*
>
> - As for the motivation of the paper, as explained in the abstract and introduction: (1) The manual ordering of tasks like UniAD could prohibit synergies between tasks. For example, planning-aware perception and game-theoretic prediction-planning are all active topics in the community. (2) The widely used BEV representation introduces challenges for long-range and long-term information processing. Thus, we aim that (1) all tasks should be able to interact with each other and (2) the spatial-temporal information extraction should be sparse. In DriveTransformer, we propose to achieve such functionality all via attention operators, for the ease of scaling and deployment.
>
>
> >**Q2: This can only be justified if the performance is significantly better than the modular architectures, however this does not seem to be the case.**
>
> We totally understand your points. First, we clarify again that DriveTransformer is a modular architecture. Second, please allow us to introduce the baselines on Table 1, 2, 3:
>
> - In Table 1 and 2, DriveTransformer performs the best among methods without expert feature distillation while achieving similar performance with ThinkTwice and DriveAdapter. **ThinkTwice and DriveAdapter are not modular architecture**. They are end-to-end methods without middle outputs like detection or online mapping, which means they do not have the benefits of easy debug and maintaining. Additioanlly, their reliance on reinforcement learning expert feature distillation makes them less applicable in reality, which is why they are separately listed by the Bench2Drive team. **In summary, DriveTransformer achieves significant improvements compared to existing modular E2E-AD systems and comparable performance with CARLA specific methods while with higher efficiency**.
>
> - As for Table 3, it has been pointed out by BEVPlaner (CVPR24) that L2 metrics are hacked in nuScenes. For example, the two works in CVPR24 (BEVPlaner and ParaDrive) have worse L2 than AD-MLP, a method without any vision input! Thus, collision rate could be a better indicator, where **DriveTransformer performs the best**. However, as discussed in BEVPlaner and Bench2Drive, closed-loop metrics might be better to assess the performance of planning and thus we  stick to closed-loop performance in most experiments.

---

> ### Author Response · Authors · 2024-11-20
>
> >**Q3: The results for detection, prediction, mapping is not shown.**
>
> Thanks for your suggestion to make the paper more comprehensive. We gave the comparison of each task with UniAD and ParaDrive in nuScenes validation set:
>
> | Method           | NDS      | mAP      | minADE   | minFDE   | MR       | IoU-Road | IoU-Lane | Latency   |
> |------------------|----------|----------|----------|----------|----------|----------|----------|-----------|
> | UniAD            | 49.8     | 38.0     | 0.72     | 1.05     | 0.15     | 0.30     | 0.67     | 663.4ms   |
> | ParaDrive        | 48.0     | 37.0     | 0.72     | -        | -        | 0.33     | 0.71     | (239.5ms)   |
> | DriveTransformer | **59.3** | **49.9** | **0.61** | **0.95** | **0.13** | **0.39** | **0.77** | **211.7ms** |
>
> where the latency of ParaDrive is obtained by calculation 663.4/2.7 (UniAD 2.77X FPS, as reported in their paper). We could observe that the proposed structure achieves better performance on middle tasks with lower latency.
>
> We add this table to the Appendix C of [our updated paper](https://openreview.net/pdf?id=M42KR4W9P5) and thanks for your advice regarding better demonstrating the advantages of DriveTransforomer.
>
>
> >**Q4: It would have useful to see the performance of ParaDrive in closed loop setting.**
>
> We think it is an important baseline too. However, ParaDrive does not open source their code yet (https://xinshuoweng.github.io/paradrive/). To complement for this, in Table 5 (a), *w/o Task-SA* is a good substitute to compare the idea of pure parallel task heads in ParaDrive or entangled ones in DriveTransformer:
>
> | Task Interaction | Driving Score | Success Rate |
> | -------- | -------- | -------- |
> | No Interaction, Pure Parallel (ParaDrive Style)     | 52.37     | 20.00     |
> | Task Self-Attention (DriveTransformer) | **60.45** | **30.00** |
>
> By comparison, we could conclude that the task level cooperation is helpful.
>
> >**Q5: This manuscript could use a grammar / spell check, there are quite a few typos.**
>
> Thanks for your advice and we have carefully . Please check the [our updated paper](https://openreview.net/pdf?id=M42KR4W9P5). We highlight our revision in blue.
>
>
> >**Q6: Task parallelism is not the right way to describe this architecture, what is parallel here?**
>
> Thanks for your advice. The term *parallel* is in opposite to the *sequential* design where there is a ordering among tasks. For example, in UniAD, TrackFormer and MapFormer are the upstream modules of MotionFormer; TrackFormer, MapFormer, and MotionFormer are the upstream modules of PlanFormer.
>
> Any advice are welcomed and we will revise our paper if there is more appropriate way to describe the structure. Thanks!
>
>
> >**Q7: What is the criterion for Dev10?**
>
> Thanks for your concerns and we will give more justifications in the main text to make it clear.
>
> For your convenience, we provide the explanation below:
> Since Bench2Drive official set has 220 routes which is exetremely expensive to run, we contact the Bench2Drive's official team and conduct series of discussions with them. The following is the justifications of the proposed Dev10:
>
>
> 1. There are 44 kinds of scenarios in Benc2Drive where there are 5 different routes under different locations and weathers for each type. Due to the low variance of Bench2Drive's short routes (as also proved by the experiments in Appendix B), selecting one route per scenario could reflect the model's ability in that case. Though the 5 runs of Bench2Drive provide results with less variance, we aim to find a more economic way. When comparing with state-of-the-art works, we stick to 220 routes to avoid overclaim.
>
> 2. Further, there are some very similar scenarios in these 44 types. For example, as shown in page 15 of Bench2Drive's paper (https://arxiv.org/pdf/2406.03877), scnenario 2 "ParkingCutIn", scenario 3 "ParkingCutIn", and scenario 4 "StaticCutIn" all examine the ability of the ego vehicle to slow down or brake for the cut-in vehicle. Thus, after discussing with the Bench2Drive official team, these scenarios could be summarized into 10 high-level types:

---

> ### Author Response · Authors · 2024-11-20
>
> - **ParkingExit**: requiring the model to drive out of a parking lot.
>
> - **ParkingCrossingPedestrian**, DynamicObjectCrossing, ControlLoss, PedestrainCrossing, VehicleTurningRoutePedestrian, VehicleTurningRoute, HardBrake, OppositeVehicleRunningRedLight, OppositeVehicleTakingPriority: requiring the model to conduct emergency brake or slow down drasticly under dangerous situations.
> - **StaticCutIn**, HighwayExit, InvadingTurn, ParkingCutIn, HighwayCutIn: requiring the model to handle cut-in behaviors of the front vehicle*
> - **HazardAtSideLane**, ParkedObstacle, Construction, Accident: requiring the model to overtake the blocking obstacles  in front of it.
> - **YieldToEmergencyVehicle**: requiring the model to give  way to emergency vehicles.
> - **ConstructionObstacleTwoWays**, ParkedObstacleTwoWays, AccidentTwoWays, VehiclesDooropenTwoWays, HazardAtSideLaneTwoWays: requiring the model to drive in the reverse lane for a short distance, complete the overtaking, and then return to the original lane.
> - **NonSignalizedJunctionLeftTurn**, SignalizedJunctionLeftTurn, InterurbanActorFlow, InterurbanAdvancedActorFlow, CrossingBicycleFlow, VinillaNonSignalizedTurn, VinillaNonSignalizedTurnEncounterStopsign, VinillaSignalizedTurnEncounterGreenLight, VinillaSignalizedTurnEncounterRedLight, TJunction: requiring the model to handle the traffic at intersections and complete its turn.
> - **BlockedIntersection**: requiring the model to yield for the blocking event within the intersection until the event is finished.
> - **SequentialLaneChange**: requiring the model to continuously change several lanes.
> - **SignalizedJunctionLeftTurnEnterFlow**, EnterActorFlows, SignalizedJunctionRightTurn, NonSignalizedJunctionRightTurn, MergerIntoSlowTraffic, MergerIntoSlowTrafficV2: requiring the model to enter the dense traffic flow and merge into it.
>
> 3. For the 10 high-level types, we select one route for each with diverse weathers and towns.
>
> 4. When comparing with other methods, we stick to 220 routes while conducting ablation studies, we use Dev10. Additionally, **Dev10 could also serve as a validation set to avoid researchers overfitting Bench2Drive220**.
>
> 5.  We will open source the Dev10 benchmark and the Bench2Drive official team plans to integrate it into their official repo to provide a short and economic validation set.
>
> Thanks for your advice to make the paper clearer and we have added the above description to the Appendix B of [our updated paper](https://openreview.net/pdf?id=M42KR4W9P5) and marked the changes in blue.

---

> > ### Comment · Reviewer_VVtW · 2024-11-25
> > **Reply to Official Comment by Authors**
> >
> > Thank you for your detailed response. I appreciate the explanation for dev 10 dataset. I have updated my rating based on your response; I think this work should be seen by the community.

---

> > > ### Author Response · Authors · 2024-11-25
> > >
> > > Thanks for your reply. We are glad that your concerns are solved and we appreciate your advice to make the manuscript better!

---

### Author Response · Authors · 2024-11-20
**General Response**

Dear AC and reviewers,

We express our gratitude to all reviewers for their valuable time and insightful comments. **We share the same excitement to explore the scaling law for E2E-AD and we will open source all our code so that members in the community could try larger models**. Following your kind advice, we added more experiments, results, and explanations to demonstrate the advantages of DriveTransformer from different perspectives. Please check [our updated paper](https://openreview.net/pdf?id=M42KR4W9P5) where the changes are marked in blue.

We hope our responses could solve your concerns. If you have any further concerns or questions to discuss, we are more than willing to address them. Looking forward to your further suggestions to improve the paper!

Thanks,

Authors of Submission 2919

---

### Author Response · Authors · 2024-11-25

Dear Reviewers,

As the deadline of author-reviewer discussion is approaching, If you have any further concerns or questions to discuss, we are more than willing to address them.

Thanks,

Authors of Submission 2919

---

### Meta-Review · Area_Chair_yjRP · 2024-12-20

**Metareview:**

This paper presents a simplified E2E-AD framework, termed DriveTransformer, to streamline end-to-end autonomous driving. The proposed method leverages attention mechanism to connect different modality, representations, and perception signals, which leads to a scalable and unified architecture. Initially, reviewers are concerned about the completeness of experiments (e.g. comparison to ParaDrive, Sparsedrive, SparseAD),  criticism on BEV Sequential Paradigm, and technical details on nuScenes. During the rebuttal period, authors provide extensive experiments to address these concerns. Reviewers are unanimously positive about this paper. Therefore, I recommend accepting this paper.

**Additional Comments On Reviewer Discussion:**

Initial concerns include: completeness of experiments (all reviewers); the individual results for detection, prediction, mapping (VVtW); and training stability of different settings (Fh94).

During the rebuttal, authors provided adequate additional experiments to address these concerns. After that, reviewers all support this paper.

---

### Decision · Program_Chairs · 2025-01-22

Accept (Poster)